# Anxiety/Depression Predominance in Liaison-Psychiatry Users of a South-East Mexico Tertiary Hospital

**DOI:** 10.3390/healthcare10071162

**Published:** 2022-06-22

**Authors:** Lizzette Gómez-de-Regil, Damaris F. Estrella-Castillo, Miguel Cicero-Ancona

**Affiliations:** 1Hospital Regional de Alta Especialidad de la Península de Yucatán, Mérida 97130, Mexico; hraeyucatan@gmail.com; 2Faculty of Medicine, School of Rehabilitation, Universidad Autónoma de Yucatán, Mérida 97000, Mexico; ecastill@correo.uady.mx

**Keywords:** anxiety, depression, psychiatric department, hospital, Mexico

## Abstract

Patients at tertiary hospitals may find themselves in need of mental health support due to the distress associated with the illness that may or not lead to a psychiatric condition. Here is an overview of the clinical cases treated by the liaison psychiatry service of a public tertiary hospital from Southeast Mexico during its first years of operation (2008–2018), with the purpose of gathering information about the status and needs of this population. A sample of 304 clinical records of patients treated for the first time by the psychiatry service was reviewed, and the distribution by demographic characteristics, diagnosis of mental illness and medical area of reference was analyzed. Anxiety and depression symptoms were the most frequent. Most patients were women, lived in Merida and returned after the first appointment. The neurology service referred most patients, yet most attended directly. General tertiary hospitals should prioritize integrating ad hoc mental and physical health care. Adult women with a profile of anxiety and/or depression would be the first target group. Some areas of opportunity for further research and improvement of mental health services are: preventive services for anxiety and depression, follow-up of patients, attention to relatives of patients at intensive care units, implementation of telehealth alternatives, training on mental health screening and inter- and intra-institutional collaboration.

## 1. Introduction

Tertiary hospitals provide services to patients whose health conditions are severe and/or chronic and, thus, require highly specialized care. These patients may find themselves in need of mental health support due to the distress associated with the illness that may or not lead to a psychiatric disorder. Although the strong link between stress and illness is widely acknowledged, the underlying mechanisms are complex [1]. Chronic stress may increase vulnerability to illness [2,3] by suppressing the immune system, releasing histamine and/or altering insulin needs [4,5]. Conversely, when facing serious illness, the patients [6,7], and even their relatives [8,9,10], may feel stressed not only about the medical prognosis per se but due to their emotional representations and perceptions of possible adverse consequences in various life dimensions (e.g., work activity, household economy, independence for functioning and long-term treatment) [11].

Improvements in medical diagnoses and treatments have increased the survival rates from illnesses that, in the past, were terminal, yet the upcoming years may come with chronic conditions that require ongoing medical resources and/or limit daily functioning. The current definition of health goes beyond merely the absence of disease or infirmity but comprises a state of complete physical, mental and social well-being [12]. Given that “there is no health without mental health” [13], in the last decades, the relevance of mental health care has increased, being well-recognized and promoted by the World Health Organization (WHO).

Caring for mental health must be present at all levels, and particularly in tertiary hospitals, where patients often arrive already emotionally exhausted and overwhelmed by illnesses that demand time, energy and economic resources. In these circumstances, patients may experience psychological distress and require mental health care for preventing or ameliorating symptoms of prolonged stress, depression and/or anxiety.

In 2012, at the 65th World Health Assembly, the WHO adopted the Mental Health Action Plan 2013–2020 (last updated at the 74th Assembly in 2021 for 2013–2030), aiming at “promoting mental well-being, preventing mental disorders, providing care, improving recovery, promoting human rights and reducing mortality, morbidity and disability of people with mental disorders” [13]. Among its objectives, the Action Plan calls for strengthening information systems, scientific data and research on mental health. Governments have a responsibility for the health of their peoples, which can be fulfilled only by the provision of adequate health and social measures, planned accordingly to the evidenced needs of their populations. Responding to this demand, this work presents an overview of the clinical cases treated by the liaison psychiatry service of a public tertiary hospital from Southeast Mexico during its first years of operation. The authors had the goal of gaining evidence of the profiles and needs of the users in order to be taken into consideration for the optimal design, implementation and evaluation of mental health services.

## 2. Materials and Methods

This exploratory and retrospective study was performed at the HRAEPY (Regional Hospital of High Specialty of the Yucatan Peninsula), a public tertiary hospital in Southeast Mexico. Both the Research and the Research Ethics Committees of the HRAEPY approved the protocol. The HRAEPY started functioning in June 2008, and even though it does not have a mental health department, during its first 11 years, up to three active staff psychiatrists were available to provide liaison services for a total of 50,863 new users.

The study targeted patients that had ever attended a psychiatry appointment at the HRAEPY. First, from the Statistics Department, the authors obtained the list of patients (recorded by name) that had received care between 2008 (the hospital’s start year) and 2018. The department in charge of the management of clinical files corroborated this list and filtered those cases that were inexistent, duplicated or had passed away. Additionally, it provided the total number of new clinical files opened each year, data to help estimate the rate of users who have ever attended an appointment at the HRAEPY’s psychiatric service. Following this, 10 files were randomly selected to corroborate that the required information would be available and, if necessary, make adjustments to the data capture sheet. The required sample size by year was estimated (Table 1), and files were randomly selected. Up to the date of the data collection, medical records were not electronically archived; thus, the clinical files were manually reviewed to gather information regarding: age, sex, dwelling location, date of first and last appointments, mental diagnosis and service referring the patient. Additionally, the total number of appointments by patient from his/her first interview to the end of 2019 was recorded. In order to prevent bias, no recorded information from 2020 and on was considered, given that the year was still ongoing during data collection and the COVID-19 pandemic demanded substantial changes in the regular services provided by the hospital. The data was analyzed with descriptive statistics.

A stratified random sampling with a population size of 1450 (original files of living patients identified by name, file number and whose names are not duplicated); a confidence level of 95% and a 5% margin error required a sample of 304 cases (Figure 1) in a range from 2 to 42 cases by year (Table 1). A total of 374 files were randomly selected in order to complete the required sample size, given that, of the initially selected files, 61 were not found in the archives, and 9 were discarded because they did not have any notes from a psychiatrist.

## 3. Results

Table 2 presents the descriptive statistics of the samples. A mean of 2.9 (1.9–3.4%) of the new cases each year attended the psychiatry service at least once. It must be remembered that not all 50,863 files ever opened at the HRAEPY were available during data collection; some might have been lost and some other currently found in the dead archives or being used by a service at the moment of request.

Out of the final sample (*n* = 304), most patients (70.7%) reported symptoms related to depression and/or anxiety disorders. In the case of the people interviewed as part of a surgery protocol, 84.0% were declared suitable. For the remaining cases, 16 files did not have any reference to a psychiatric symptom or disorder. Most of the cases (29.6%) attended the psychiatry service as their first appointment, whereas most medical referrals came from the neurology service (21.1%). As part of their protocols, the bariatric and the transplant surgery services referred 6.6% and 5.6% cases, respectively.

Most patients were females, and the mean age at the time of their first appointment was 48.0 years, with a range from 17.3 to 92.1 and no significant differences by sex (*p* = 0.25). Most patients lived in the State of Yucatan (88.2%), either in the city of Merida (40.5%), where the hospital is located, or in adjacent locations (11.5%).

Regarding the time from first to last appointment with the psychiatry service, the average was 19.4 (SD = 26.7) months. When excluding the cases that attended only once (21.1%), the average was 24.6 (SD = 27.9) months, ranging from 2 days to 120 months. During their first year, 36.5% of the patients returned but dropped out, 13.8% during the second year and 11.5% during the third year, and only 17.1% returned after that time. Yet, it could be the case that a patient interrupted treatment for some time and later retuned. Most patients (78.9%) returned after their first appointment, with 38.8% having between 2 and 5 visits, 17.8% between 6 and 10 visits and 22.8% between 11 and up to 55 visits. The average number of visits was 7.4 (SD = 8.6), ranging from 1 to 55. Here, it must be considered that the older the file, the more chances the case added follow-up appointments. Moreover, it is very likely that the physical files of those patients that had not returned for some time had already been removed from the clinical archive at some depuration point.

## 4. Discussion

An important number of people with long-term physical health conditions also have mental health problems; this comorbidity can severely impact health outcomes and reduce the quality of life [1,14]. For instance, depression is twice as common in people with diabetes relative to the general population, and in the other direction, comorbid depression in diabetes has an additive negative effect on patients’ mental quality of life and is associated with an increased risk of debilitating complications, further increasing the disease burden [15]. It is estimated that up to 18% of the amount expended on long-term conditions is in relation to deteriorated mental health. By interacting with and exacerbating physical illnesses, comorbid mental health problems may raise the total health care costs by at least 45%. Innovative forms of liaison psychiatry demonstrate that providing better support for comorbid mental health needs can reduce physical health care costs [1].

At tertiary hospitals, most patients arrive by medical reference and dealt with their illnesses for some time. It is not unusual that they experience burden and despair that may lead to the manifestation of incipient or full-blown mental disorders. Learning about the main features and needs of patients is a mandatory step towards the design, implementation and validation of strategic health programs.

At the HRAEPY, most patients attending its psychiatric service are women in their late-middle adulthood years. This may reflect that females are not only more prone to present mental/emotional symptoms [16,17,18] but also to seek treatment earlier [17,18,19]. It is important to provide early intervention in mental health to these users, not only to prevent a mental disorder and the associated economic costs of a comorbidity but also considering that, in Mexican culture, women, at home full-time or not, are often the family’s primary caregivers, so their emotional well-being would have a significant effect on their family. Lara and colleagues, considering social and gender factors, developed psycho-educational interventions for women with depressive symptoms [20] to prevent women from developing postpartum depression [21] and the Help for Depression (HDep) program [22,23], one of the few unguided web-based interventions available in Latin America, all with promising results. Regarding age, it is important to notice that the HRAEPY does not provide pediatric care (except for the Pediatric Cardiac Intensive Care Unit), so all patients attending the psychiatric service are adults. However, the HRAEPY must consider that every Pediatric Intensive Care Unit should be able to address not only the physical but also the psychosocial, emotional and spiritual needs of patients with life-threatening conditions and their immediate families [24].

Most patients of the psychiatric service presented symptoms of anxiety and/or depression. These mental disorders are not only the most prevalent in patients with chronic illnesses [6,25,26,27] but also in the general population [16,28]. It must be underlined that presence of “symptoms of” does not equal the “clinical diagnoses of”. Patients who experience mild symptoms of anxiety and/or depression (an expected response when facing a severe, chronic and/or disabling physical illness) may better benefit from preventive services (e.g., counseling, group therapy, psycho-education and support groups) to help them manage their cognitions and emotions and increase adherence to treatment [2,29,30]. This highlights the importance of expanding efforts to enable society to identify anxiety and depressive symptoms as obstacles to well-being and productivity and to accept them as challenges that can and must be overcome with the decisive and organized participation of everyone.

Most patients returned after their first appointment; yet, it must be considered that the ones that had not returned for a long time might have already been taken out of the archive. The patient might have dropped off treatment due to the amelioration of symptoms or receiving care somewhere else. However, on the contrary, it might be the case that the patient did not continue under treatment due to limitations in accessing care (e.g., economic and geographical), a lack of insight regarding mental illness or even death. Research on health trajectories [31,32,33] comes forward as an ideal model to monitor the routes of care in mental health and chronic diseases, learn about the likely limitations for access and improve health care provisions. 

With most patients living in the hospital’s city, it would be feasible to provide in-person individual and/or psychoeducational group interventions to develop self-management of health. These interventions must be designed accordingly to the target group, taking into consideration not only clinical features but also psychological and social factors that may intervene [34]. In Mexican culture, there is a marked influence of a male-dominated society and the stigmatized concept of mental disorders. Within the indigenous Mayan population (mostly located at the Yucatan Peninsula), the traditional healers provide a remedy for a wide range of mental health problems, so it is important to reach a compromise that include the individual’s cosmovision (worldview) [35]. In addition to this, Yucatan is positioned in the first places in alcoholism nationwide, where the rates of suicides and alcohol intoxications are among the leading causes of mortality [36]. Patients may not report the use of alcohol and substances, either from shame and/or as it is normalized in the community and not even seen as a risk for physical and mental health.

Patients with limitations to attend in-person appointments (e.g., due to distanced dwelling, mobility impairments and/or limited schedule) may benefit from alternative provision of services (not yet available at the HRAEPY). Telehealth allows the delivery of universal health coverage through technology in order to provide quality and cost-effective health services and eliminate the need for either the patient or practitioner to travel to appointments, offering potential help in reducing the ‘mental health gap’ in low- and middle-income countries [37,38]. Although remote therapy offers a number of advantages, it brings about a variety of challenges that are unique to this modality. For example, pre-recorded video modules are less interactive with the patient, and teleconferences in real time require trained staff for technological support. Partial or total contents of online interventions in mental health may be accessed 24/7, and research has shown favorable results in the clinical [37,39] and economic [40] outcomes. Yet, this option is not always suitable. From the age, education and socioeconomic levels of most people attended to at the HRAEPY, it can be assumed that they are unlikely to be computer literate and own an electronic device with reliable internet access. Moreover, an important segment of users only speak (but not write) Maya, the local indigenous language rarely known by health professionals. The use of telehealth is expected to increase as the upcoming generations of patients and health professionals are more skilled in the use of technology and internet access spreads, so further research and training will be needed to provide an effective and more widely distributed low-cost mental health approach, a task that the HRAEPY must adopt.

At the HRAEPY, the psychiatric service does not offer hospitalization and functions mainly as a support to patients treated by other medical specialities; thus, intra-institutional referrals are predominant. Despite the benefits, referral to consultation–liaison psychiatry remains low; a thoughtful analysis would help clarify whether this is due to low rates of psychiatric symptoms or to non-detection and/or nonreferrals from other specialists. Education should be provided to hospital doctors but also to nurses, residents and social workers to better recognize mental illness so referrals can see an increase [41,42]. Collaborative screening and accurate referrals would require the consolidation of a multi- and interdisciplinary mental health service [43,44,45].

From clinical records, it was observed that most of the patients directly sought the psychiatry services; it would be worth exploring the source of this informal referral, how patients acknowledged their need for mental health care, how they reached for treatment at HRAEPY and whether or not they had previously received psychiatric treatment. Among the medical departments at the HRAEPY, the one with the most percentage of patient referrals to psychiatry was the neurology department. Although this study does not count with the specific diagnosis, it does concur with previous findings [43,46,47], where the neurology team stands out as the highest number of referrals to psychiatry, mainly in relation to brain organic disorders, mood and dissociative disorders.

At the HRAEPY, a psychosocial evaluation is mandatory for all patients before transplant or bariatric or surgery; yet, it seems to be limited to meeting the requirement, as no follow-up appointments are routinely scheduled. The psychological care for transplant recipients and donors should continue throughout the postoperative period to guarantee treatment adherence and a successful process [48]. In the case of bariatric patients, an evaluation is recommended to identify potential contraindications to surgical intervention or barriers for lifestyle changes [49]. The published recommendations point out that the optimal frequency for follow-up visits for bariatric patients are seven or more visits per year [50].

The data collection was limited by the COVID-19 pandemic, where all recorded information (if any) regarding the psychiatric service from 2020 and on was not included to prevent bias. Populations were exposed to prolonged uncertainty and lockdown, and this might have had a significant effect on mental health, with an increment in rates of symptoms of depression and/or anxiety and an exacerbation of illness in psychiatric patients. In England, it was estimated that up to 30% of people in the general population experienced trajectories with symptoms in the clinical range during the lockdown; yet, the trajectory patterns were diverse [33]. Depressive and anxiety disorders were the leading causes of the global health-related burden in the years prior to the pandemic, but the social restrictions created an environment where psychological distress and mental illness were likely exacerbated. The COVID-19 pandemic has created an increased urgency to strengthen mental health systems, demanding the early detection and management of mental and emotional health issues for managing the global prevalence and burden of depressive and anxiety disorders in the upcoming populations of patients [51].

This study follows a simple design; however, the value of this research relies, above all, on the utility of the analysis and the feasibility of replicating the method in other hospitals from the same hospital network. At the moment, across Mexico, there are five HRAEs (Regional Hospitals of High Specialization). Apart from being tertiary hospitals dependent on the Federal Government, they also have in common the task of performing up-to-date research in benefit of their populations, and this study would be easy to replicate across institutions alike, in Mexico and/or abroad.

Given the inner conditions for research at the site, the study followed a descriptive/observational design. The results came from information available in nonelectronic files, which was not only scarce but, in many cases, missing. This natural nonmanipulated condition, evidencing the usual functioning of the services, implies that a deeper analysis could not proceed without questioning the validity and reliability of data for a generalization of the findings. Further projects must consider the standardized inclusion of other relevant variables for more precise characterizations of users. For instance, coded diagnoses (CIE and/or DSM) would help establish whether symptoms are present at the subclinical or clinical level and/or in comorbidity with a physical illness, which increases the economic [52,53] and psychological costs [1,14]. Additionally, they would reflect more accurately the evolution of the patient’s health status.

Lastly, the implementation of electronic files [54,55,56] would standardize the recording of relevant information for follow-up (e.g., date, updated diagnosis and prescription) from all services each patient attends and would also reduce discrepancies between Statistics and Archive departments’ information. Given that the descriptive statistics reported in this study were based on only 304 samples, all findings and implications are not conclusive.

## 5. Conclusions

General tertiary hospitals should prioritize integrating ad hoc mental and physical health care more closely as a fundamental strategy for improving the quality and functionality of their provision of health care. Adult women with a profile of anxiety and/or depression would be the first target group to attend. Some areas of opportunity for further research and improvement of mental health services are: preventive services for anxiety and depression, follow-up of patients, attention to relatives of patients at intensive care units, the implementation of telehealth alternatives, training in mental health screening and inter- and intra-institutional collaborations.

## Figures and Tables

**Figure 1 healthcare-10-01162-f001:**
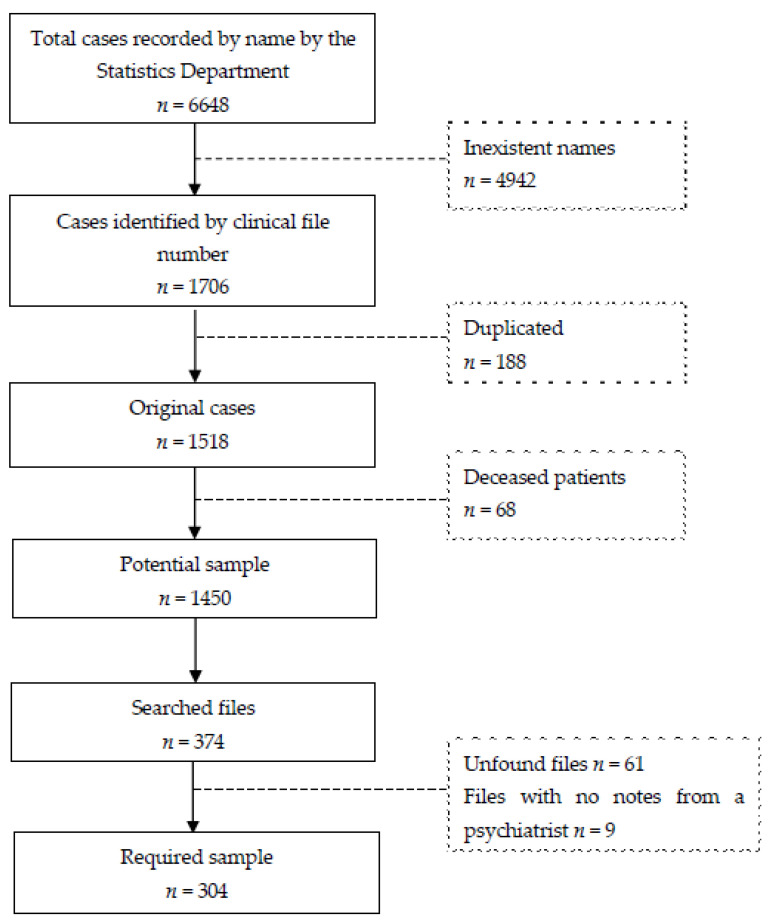
Sample flow diagram.

**Table 1 healthcare-10-01162-t001:** Estimation of the sample size.

	Total New Files (*n*)	Potential Cases * (*n*/%)	Distribution (%)	Required Sample (*n*)
2008	455	11 (2.418)	0.758	2
2009	4253	81 (1.905)	5.582	17
2010	5487	152 (2.770)	10.476	32
2011	6401	199 (3.109)	13.784	42
2012	5619	170 (3.025)	11.716	36
2013	3964	136 (3.431)	9.373	29
2014	6400	190 (2.969)	13.094	40
2015	5209	126 (2.419)	8.684	26
2016	4686	149 (3.180)	10.269	31
2017	4432	130 (2.933)	8.959	27
2018	3957	106 (2.679)	7.305	22
Total	50,863	1450 (2.851)	100	304

* Original files of patients identified by name, file number and whose names are not duplicated and are not registered as deceased. Percentage by total new cases each year.

**Table 2 healthcare-10-01162-t002:** Descriptive statistics.

**Distribution by Year**												
Year of file		2008	2009	2010	2011	2012	2013	2014	2015	2016	2017	2018
New files (*n* = 50,863) *n*		455	4253	5487	6401	5619	3964	6400	5209	4686	4432	3957
Potential cases * (*n* = 1450) *n*		11	81	152	199	170	136	190	126	149	130	106
% over total yearly new cases		(2.4)	(1.9)	(2.8)	(3.1)	(3.0)	(3.4)	(3.0)	(2.4)	(3.2)	(2.9)	(2.7)
Final sample (*n* = 304) *n*		2	17	32	42	36	29	40	26	31	27	22
Time (months) from first to last contact (*n* = 304)												
Minimum		33.0	0.0	0.0	0.0	0.0	0.0	0.0	0.0	0.0	0.0	0.0
Maximum		93.1	120.4	116.3	108.1	73.9	79.7	67.5	53.1	41.5	35.3	19.7
Mean		63.1	47.5	31.6	27.6	18.7	15	11.7	17.4	12	11.1	4.9
Standard Deviation		42.5	41.0	35.4	36.2	20.6	22.5	19.3	16.2	13.3	10.7	6.2
Total of appointments (*n* = 304)												
Minimum		21	1	1	1	1	1	1	1	1	1	1
Maximum		36	55	36	50	25	33	25	32	25	17	8
Mean		28.5	12.2	9.8	9.2	6.9	6.4	5.1	8.2	6.3	5.5	3.6
Standard Deviation		10.6	13.2	10.2	12.2	6	8.1	6.3	7.4	6.9	4.3	2.4
**Characteristics of sample (*n* = 304)**												
Diagnosis (*n*/%)												
Mixedanxiety/depression	81 (26.6)	Surgery protocol				25 (8.2)	Psychosis				10 (3.3)	
Anxiety	79 (26.0)	No diagnosis/symptoms				16 (5.3)	Substance abuse				6 (2.0)	
Depression	55 (18.1)	Dementia				12 (3.9)	Other				20 (6.6)	
Referring service (*n*/%)												
Psychiatry	90 (29.6)	Internal medicine				30 (9.9)	Transplant protocol				10 (3.3)	
Neurology	64 (21.1)	Emergency admission				20 (6.6)	Gastroenterology				10 (3.3)	
Surgery	56 (18.4)	Bariatric surgery				17 (5.6)	Other				7 (2.3)	
Sex (*n*/%)		Age (years at the time of first interview)					Dwelling location (*n*/%)					
Female	219 (72.0)	Minimum				17.3	Yucatan				268 (88.2)	
Male	85 (28.0)	Maximum				92.1	Quintana Roo Campeche				19 (6.3)	
		Mean				48	Other				12 (3.9)	
		Standard Deviation				16.8					5 (1.6)	

* Original files of patients identified by name, file number and whose names are not duplicated and are not registered as deceased. Percentage by total new cases each year.

## Data Availability

The dataset generated and/or analyzed during the current study is available from the corresponding author upon reasonable request.

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
