# Peer review of "Anxiety/Depression Predominance in Liaison-Psychiatry Users of a South-East Mexico Tertiary Hospital"

_healthcare, 2022, doi:10.3390/healthcare10071162_

Round 1

Reviewer 1 Report

The study called "Prevalence of anxiety/depression in users of liaison psychiatry 2 of a third-level hospital in southeastern Mexico" is considered of great interest to readers. However, the theoretical content is brief and the main axis of the study topic should be investigated in a reinforcement of the content of the literature,
since which is weak in its scientific soundness. Regarding the section of methods as results, it is quite weak, in which the authors only apply a descriptive analysis, being a limited study to specify data that facilitate the understanding of the investigated topic and allows conclusive conclusions to be reached. For this reason, the discussion as a conclusion needs to be rewritten with the intention of giving a more consistent framework of the results obtained, with the application of new scientific methods that provide more rigor to their study.

Reviewer 2 Report

The study presents a summary of patient characteristics from a tertiary hospital, but does not address any deeper analysis.

The prevalence of anxiety/depression problems in psychiatric diagnoses or their relationship with the used medical services will be of interest for the internal management and planning of the hospital, but the interest for the general public is not highlighted.

The undoubted work involved in obtaining sample information from medical records constitutes a first step to address a more specific research objective, but, by itself, it does not justify a research article.

Reviewer 3 Report

It was a great pleasure that I reviewed the manuscript entitled “Anxiety/depression predominance in liaison-psychiatry users of a South-East Mexico tertiary hospital.” I think the paper presents some findings that might be of interest for some readers. Here are several comments and suggestions that might help strengthen the paper. I am presenting those comments and suggestions in (mostly) chronological order.

1.     I think the authors should state more specific purpose or goal of the study in Abstract.

2.     Similarly, the goal or purpose of the study is not clear in the Introduction section, although it is well-written providing the reason why the authors conducted this study and overviewing the background information related to this topic.

3.     The authors mentioned that Appendix 1 provides how they determined the required sample size of this study. I would suggest they should provide in the Materials and Methods section, because it is crucial information for this type of study.

4.     More generally, I was surprised that the authors needed to review the clinical files manually from 2008 to 2018. When did the Mexican authorities start recording the information in electronic files?

5.     I think the first and second paragraph of the Results section should be moved in the Materials and Methods section because they were still discussing how the samples were selected.

6.     I had trouble understanding the lines from 141 to 143 starting from “Nevertheless…” Please consider rephrasing.

7.     I think it is important to clarify that what the authors reported from the third paragraph of the Result section (i.e., lines from 144) were based on 304 samples.

8.     Overall, the discussion section is well-written. However, as related to the above point, given that descriptive stats reported were based on only 304 samples (albeit I understood the difficulty of coding), I think it is important to address all findings and implications the authors stated based on those findings were not quite conclusive.

Round 2

Reviewer 1 Report

After the clarifications and changes made by the authors following the instructions of the reviewers, the article has taken on more strength and coherence, for which the article is considered publishable.

Reviewer 2 Report

The changes made to the paper clarify the objective and results of the work, addressing the deficits noted above. The current version of the article correctly shows the organizational needs of a tertiary hospital and is useful for professionals and researchers interested in this topic.

Reviewer 3 Report

Thank you for addressing my comments.